# Effect of Temperature and Humidity on Milk Urea Nitrogen Concentration

**DOI:** 10.3390/ani13020295

**Published:** 2023-01-14

**Authors:** Takula Tshuma, Geoffrey Fosgate, Edward Webb, Corlia Swanepoel, Dietmar Holm

**Affiliations:** 1Department of Production Animal Studies, Faculty of Veterinary Science, University of Pretoria, Private Bag X 04, Pretoria 0110, South Africa; 2Department of Animal Science, Faculty of Natural & Agricultural Sciences, University of Pretoria, Private Bag X 20, Pretoria 0028, South Africa; 3Hatfield Experimental Farm, University of Pretoria, Private Bag X 20, Pretoria 0028, South Africa

**Keywords:** variation, urea nitrogen concentration, environmental factors, Holstein, cows

## Abstract

**Simple Summary:**

Milk urea nitrogen concentration varies between and within cattle breeds. The variation is thought to be mainly influenced by dietary protein intake. For this reason, proposals have been made to utilize milk urea nitrogen concentration to monitor the protein nutrition of cattle. The impact of environmental factors on milk urea nitrogen concentration is unknown. Extreme concentrations (too low or too high) of milk urea nitrogen are associated with poor reproductive performance in cows. This study investigated the effect of ambient temperature and humidity on milk urea nitrogen concentration of cows. Temperature and humidity on the day of milk sampling were positively associated with milk urea nitrogen concentration. Temperature had a significant influence on measured urea nitrogen concentration, and for this reason, it should always be considered and accounted for when milk urea nitrogen concentration data are used to make inferences about the dietary management of cows or when identifying cows that might be at risk of poor reproductive performance caused by having extreme milk urea nitrogen concentrations. Different rations should be formulated for winter and summer to keep the milk urea nitrogen concentrations within acceptable limits.

**Abstract:**

This study investigated the effect of ambient temperature and humidity on milk urea nitrogen (MUN) concentration in Holstein cows. Meteorological data corresponding to the dates of milk sampling were collected over six years. A linear mixed-effects model including a random effect term for cow identification was used to assess whether temperature and humidity were predictive of MUN concentration. Age, days in milk, temperature humidity index (THI), ration, milk yield, parity and somatic cell count were also evaluated as main effects in the model. A general linear model including all variables as random effects was then fitted to assess the contribution of each variable towards the variability in MUN concentration. Maximum daily temperature and humidity on the sampling day were positively associated with MUN concentration, but their interaction term was negatively associated, indicating that their effects were not independent and additive. Variables that contributed the most to the variability of MUN concentration were dietary crude protein (21%), temperature (18%) and other factors (24%) that were not assessed in the model (error term). Temperature has a significant influence on urea nitrogen concentration and should therefore always be considered when urea nitrogen concentration data are used to make inferences about the dietary management of dairy cows.

## 1. Introduction

Milk urea nitrogen (MUN) and blood urea nitrogen (BUN) concentrations in cattle have been well studied [1,2] because of their important association with reproductive performance [3,4,5]. Whether MUN or BUN concentration is measured in a study should not affect the conclusions thereof, because urea nitrogen is known to equilibrate rapidly between body fluids, hence the existence of a close relationship between MUN and BUN concentrations. MUN concentration can be estimated from BUN and vice versa [6,7].

Urea nitrogen concentration in blood or milk varies between and within cattle breeds [3,8,9,10,11]. It is also known to vary from herd to herd and from cow to cow within the same herd [3,12]. This variation is thought to be mainly influenced by the level and type of dietary protein supplementation [13,14], although other factors are known to play a role [3,15,16,17]. At times, the reported effects are conflicting, and the direction of the effects is not fully understood. For example, some studies reported that breed significantly affected BUN concentration [3,15,18], whereas others found no effect [16,19]. Another example is the effect of cow age on MUN concentration. Heifers were reported to have lower MUN concentration than older cows [20], whereas other studies identified no association at all [21]. To the knowledge of the authors, no studies have investigated the effects of ambient temperature and relative humidity on MUN concentration.

If MUN or BUN concentration data are to be successfully used to monitor dietary and management factors of dairy cattle [22] or to identify cows that are at risk of having reduced reproductive performance because of having extreme urea nitrogen concentrations [15], then all factors that influence the measured concentration under field conditions must be clearly understood.

The main objective of this study was to determine how environmental factors (temperature and relative humidity) affect MUN concentration and to estimate the percentage contribution of these factors to the variability in measured MUN concentration.

## 2. Materials and Methods

This retrospective cohort study used herd data from the University of Pretoria Hatfield Experimental Farm (located in the Gauteng Province of South Africa), which were collected over six years (February 2012 to November 2017). Each milk sampling event per cow served as the unit of analysis.

### 2.1. Data Collection and Animal Management

Data from 161 high-producing Holstein cows were collected during the study period. The age of the cows ranged from 21 to 90 months and the parity ranged from 1 to 5. Cows were intensively managed in one group, milked three times a day, and fed a total mixed ration (TMR) that was delivered thrice daily. Cows received commercially prepared lactation diets from three feed companies during the study (Diet 1, 2 and 3). Diet 1 was fed from February to December 2012. This was followed by Diet 2, which was fed from January 2013 to December 2014 and also from February 2017 to December 2017. Diet 3 was fed from January 2015 to December 2015. Composite milk samples (*n* = 2174) were collected from cows every five weeks for the determination of MUN concentration and somatic cell count (SCC) by a private laboratory (South African Stud Book Association, Pretoria, South Africa). Milk sampling was performed aseptically by trained technicians from the laboratory during the mid-day (12h00) milking session. Milk samples were collected into 35 mL plastic tubes containing the preservative bronopol and transported at environmental temperature to the private laboratory within 6 h. Testing for MUN concentration was done by infrared spectroscopy (FOSS Analytical, Hillerød, Denmark). SCC was simultaneously analyzed by the same machine. The age of the cows (in days), days in milk (DIM), milk production (during the past 24 h), MUN concentration and parity data for each test date were obtained from the farm’s electronic records system (S.A.E. Afikim, Kibbutz Afikim, Israel).

Meteorological data were obtained from the nearest (less than 1 km away) South African Weather Services (SAWS) station and consisted of daily minimum and maximum temperature (°C), and daily relative humidity (%), measured three times a day (at 08h00, 14h00 and 20h00). Temperature and relative humidity were defined as the maximum value recorded for the day. The temperature humidity index (THI) was calculated as follows [23]:THI = 0.8 × T + RH/100 × (T − 14.4) + 46.4
where T = maximum daily ambient temperature in °C and RH = maximum daily relative humidity.

Daily THI values were categorized into three levels of heat stress [23,24] as follows: Low (THI ≤ 72), Moderate (72 < THI < 78) and Severe (THI ≥ 78), and these categories were used in the statistical analyses.

### 2.2. Statistical Analyses

The normality of continuous data was assessed by plotting histograms and performing the Shapiro–Wilk normality test. Data satisfying the normality assumption were reported as mean ± standard deviation (SD) and non-normal data were presented as the median and range (minimum, maximum). Non-normal variables were log-transformed prior to statistical analysis whereas normal data were entered into the model unchanged. The relationship between variables and MUN concentration was descriptively evaluated using scatter plots before statistical modelling.

A linear mixed-effects model including a random effect term for cow identification was used to assess whether the ambient temperature and relative humidity were predictive of MUN concentration. Age, CP, DIM, fat, THI, ration, milk conductivity, NFC, previous 24 h milk production, parity, and SCC were also evaluated as main effects in the model. The interaction term between relative humidity and temperature was also evaluated. Variables were initially screened individually and then incorporated into a multivariable model if they were significant at *p* < 0.2 (2-sided). Non-significant (*p* > 0.05) predictors of MUN concentration were removed one-by-one from the multivariable model in a backwards elimination process. Important confounding variables, which caused a 15% or greater change in estimates for other covariates when removed from the model, were retained.

The contribution of each variable towards the variation in MUN concentration was evaluated using a general linear model where all variables were entered as random factors (variance components analysis).

All data were analyzed using SPSS (IBM SPSS Statistics Version 23; International Business Machines Corp., Armonk, NY, USA), and the significance threshold was set at *p* < 0.05.

## 3. Results

Two thousand one hundred and seventy-five composite milk samples were collected from 161 cows during the study period. Cows produced an average of 31.4 ± 7.4 kg of milk per day, with a mean MUN concentration of 13.67 ± 4.21 mg/dL and a geometric mean SCC of 5.1 ± 0.58 per mL (Table 1). Mean temperature and relative humidity were 26.09 ± 5.05 °C and 64.45 ± 19.12%, respectively. The age of the cows at the time of sampling ranged from 21 to 90 months. Half of the cows (54%) were followed only into the second parity, whereas 27%, 7% and 2% were present in the herd up to the third, fourth and fifth parity, respectively.

Nine of the 14 variables that remained after individual screening were significant predictors of MUN concentration and remained in the regression model (Table 2). Maximum temperature and maximum relative humidity on the day of sampling were positively associated with MUN concentration (*p* < 0.001), whereas the interaction term between these variables was negatively associated with MUN concentration (*p* < 0.001). Crude protein (*p* < 0.001), NFC (*p* < 0.001), fat (*p* < 0.001), previous 24 h milk yield (*p* = 0.011) and DIM (*p* < 0.001) were all positively associated with MUN concentration, whereas SCC (*p* < 0.006) was negatively associated.

Variables that contributed the most to the variability of MUN concentration in this study were the dietary CP (21%), temperature (18%), and unmeasured variables (24%) as reflected in the error term of the model (Table 3).

## 4. Discussion

The primary objective of this study was to estimate the influence of ambient temperature and relative humidity on the variability of MUN concentrations in dairy cattle. The magnitude of the influence of these factors was also evaluated to identify those factors that should be accounted for whenever urea nitrogen concentration data are used in monitoring protein nutritional status [22] of cattle or identifying cows that might be at risk of poor reproductive performance [15].

The mean MUN concentration observed in this study was within the typical limits of between 10 and 14 mg/dL for dairy cattle [25,26], indicating that cows had an ideal nitrogen utilization efficiency [27]. Mean daily milk yield was similar to that of other high-producing herds [28], but the mean SCC was higher than the recommended cut-off point of less than 250 × 1000 cells/mL [29]. Our SCC measurements had a large standard deviation, most likely owing to the sampling of cows with either subclinical or clinical mastitis. Such cows were kept in the dataset because most cows on South African dairy farms do experience mastitis (subclinical or clinical) at some stage during their lifetime. These cows get treated, and upon recovery, they re-join the milking herd. Including these cows in the analysis makes the findings of this study relevant to typical dairy farms. Mean maximum temperatures and humidity were typical of humid subtropical climates [30]. 

Similar to other studies [1,12,25], we identified a positive association between milk yield and MUN concentration. The positive association observed in these studies was likely caused by feeding strategies aimed at providing more protein and energy to high-producing cows. Energy is known to be the main limiting nutrient in high-producing dairy cows [31]. In our study, where all cows were fed the same diet regardless of milk yield, high-producing animals likely increased their dry matter intake (DMI) in an attempt to meet their energy requirements. In so doing, they might have consumed excess protein, which likely explains the positive association between MUN concentration and milk yield. However, the contribution of milk yield to the variability of MUN concentration was relatively small (3%), suggesting that this factor could be ignored when using MUN concentration to make inferences about the dietary management of cows and their risk for reduced reproductive performance.

The negative association between SCC and MUN concentration observed in this study is similar to earlier reports [25,32], although other studies reported no association between SCC and MUN concentration [21]. We have no explanation for this negative association, but cows with higher SCC are more likely to have clinical mastitis. It is possible that these cows were under stress and stressed cows eat and drink less, leading to dehydration and a negative energy balance (NEB). The negative association between hydration status [4,17] and NEB [33,34] with urea nitrogen concentration has been previously described. Nonetheless, SCC’s contribution to the variability in MUN concentration was also small (7%), suggesting less importance when interpreting MUN concentration data.

MUN concentration increased as the DIM progressed, which is similar to other reports [26,35]. However, herds in those previous reports were fed diets containing higher protein as lactation progressed. Because of this management style, it was not possible to distinguish whether the higher MUN concentration was caused by the increased DIM or higher levels of dietary protein. Protein levels in our study were kept the same as DIM progressed, except for the unintentional variation caused by seasonal fluctuation in the quality of raw materials such as Lucerne. This led us to believe that the reason for the observed higher MUN concentrations was associated with increased DIM and not the diet. The contribution of DIM to the variability of MUN concentration was also small (8%) and the authors believe that it might not be important to consider when interpreting MUN concentration data. 

As expected, dietary CP was positively associated with MUN concentration [13,14]. Surprisingly, the association between dietary NFC with MUN concentration was negative, contrary to previous reports [36,37]. The levels of NFC in our study were within typical ranges for dairy cattle of 32 to 42% of dietary DM [37]. It is possible that some unmeasured aspect of the diet could have influenced this relationship. Most likely, the DMI of individual cows could provide some clues. However, this field study was performed under typical South African farming conditions, using real commercial TMRs where the exact nitrogen intake of each cow was not known. It was therefore not possible to analyse the CP intake. The impact of a TMR on MUN concentrations is often judged under these conditions where the intake of individual cows is unknown.

The contribution of CP to the variability of MUN concentration reported in our study is lower than the 69.3% reported in another study [38]. This difference might be explained by the fact that our study did not only focus on nutritional factors, but also included environmental factors as well. There is consensus that diet is the major predictor of MUN concentration. However, the findings of the current study suggest that within commercially prepared dairy rations, the contribution of diet is similar to that of environmental factors. This finding is likely because commercial dairy rations are specifically formulated to meet the dietary needs of a dairy cow.

Temperature and humidity were both positively associated with MUN concentration. We had expected a negative association between humidity and MUN concentration because of reduced loss of cellular fluid if the atmosphere is very humid. This would have been consistent with an earlier study, which reported that BUN concentration was lower during the rainy season [39], and yet it was similar during winter and summer. However, the climate where our study was performed is such that high humidity is usually accompanied by high temperatures. It is therefore likely that temperature and humidity did not act independently in influencing MUN concentration. This was confirmed by the negative association between the interaction term (temperature × humidity) and MUN concentration, whereas the individual variables were both positively associated with MUN concentration. This observation led us to believe that our calculated THI values were inferior indicators of heat stress compared to including the interaction term. We assumed that the effects of exposure to high temperature and humidity on MUN concentration occur immediately. However, there is still a possibility that MUN concentration that is measured in the morning might be a result of exposure that occurred many hours before measurement. The study did not investigate the possibility of a 24 h lag between the effect of environmental conditions on MUN concentration, and this is a limitation of the current results.

Our data also cannot explain the causal pathway for the association between heat stress and MUN concentration. However, we hypothesize that the effect of heat stress on MUN concentration could be explained by the dehydration that occurs in heat-stressed animals [40]. Dehydration leads to elevated BUN and MUN concentrations by regulating urea reabsorption by the kidney [41,42]. Expression of urea transporter-A (UT-A) mRNA was significantly higher in the inner medulla of dehydrated rats compared to controls [43]. Urea contributes to the osmolarity of the filtrate during reabsorption of water in the kidney. The increased expression of UT-A mRNA in dehydrated rats might have indicated an increase in urea reabsorption as a means to raise renal osmotic pressure, leading to increased reabsorption of water, and simultaneously, an increase in BUN concentration [43].

The relatively large contribution of temperature (18%) and relative humidity (9%) towards the variability in MUN concentration in this study suggests that using MUN concentration data alone to make inferences about the diet might lead to inaccurate conclusions. We suggest that temperature and relative humidity on the sampling day be considered in the interpretation of MUN data before making any adjustments to the diet. This is an important finding because MUN concentration above 19 mg/dL [44] is known to negatively affect the fertility of cattle, regardless of whether the extreme MUN concentration is due to dietary or environmental effects. From this observation, we hypothesize that the well-known negative association between heat stress and fertility of cattle [45] might be at least partly explained by elevated MUN concentrations caused by high temperature and relative humidity. More research is required to evaluate whether heat stress acts via MUN concentration or acts directly on cow fertility.

We studied one herd, which was conveniently located close to the SAWS station to minimize discrepancies in temperature and humidity readings between the station and the herd. Although the station was very close (less than 1 km) to the herd, there is still a possibility that temperature and humidity readings at the station were not representative of the microclimate experienced by the herd. However, if any such differences occurred, they should not have been substantial. The use of a single herd limits the generalizability of our findings to other dairy herds with different management practices. We do not know if other cow breeds or herds would have shown a similar magnitude in the effect of ambient temperature and relative humidity on the variability of MUN concentration.

Despite these limitations, demonstration of a significant effect of maximum ambient temperature and relative humidity on the variability of MUN concentration is an important finding that we expect to be robust.

## 5. Conclusions

Maximum daily ambient temperature and relative humidity had a significant influence on measured urea nitrogen concentration, and for this reason, they should be considered when urea nitrogen concentration data are used to make inferences about the dietary management of cows. Additionally, herd managers that utilize previous urea concentration measurements to identify cows that might be at risk of poor reproductive performance should adjust those measurements in line with the ambient temperature and relative humidity, for the data to be correctly interpreted.

## Figures and Tables

**Table 1 animals-13-00295-t001:** Descriptive statistics (means and medians with standard deviations and ranges in parentheses) for 161 Holstein cows enrolled in the study to investigate the effect of ambient temperature and relative humidity on milk urea nitrogen (MUN) concentration. Data were collected from the Experimental Farm, University of Pretoria, during the period from February 2012 to November 2017.

Variable	*n*	Mean (SD)	Median (Min, Max)
MUN (mg/dL)	2174	13.67 (4.21)	
Relative humidity (%)	2171	64.45 (19.12)	
NFC (%)	2174	41.71 (2.57)	
Milk yield (kg/day)	2172	31.4 (7.4)	
Temperature (°C)	2171	26.09 (5.05)	
CP (%)	2174	15.52 (0.42)	
Fat (%)	2174	5.73 (0.23)	
log SCC	2174	5.1 (0.58)	
SCC × 100	2174		100 (6, 9730)
DIM (days)	2162		166 (1, 877)

MUN—milk urea nitrogen; NFC—non-fiber carbohydrates; CP—crude protein; log SCC—geometric mean of the somatic cell count; min—minimum; max—maximum; DIM—days in milk.

**Table 2 animals-13-00295-t002:** Predictors of MUN concentration (from a multivariable linear mixed-effects regression model) in the study to investigate the effect of ambient temperature and relative humidity on milk urea nitrogen (MUN) concentration. Data were collected from 161 Holstein cows at the Experimental Farm, University of Pretoria, during the period from February 2012 to November 2017.

Variable	$\hat{\beta }$	SE	95% CI of $\hat{\beta }$		*p* Value
			Lower	Upper	
CP %	20.666	1.981	16.782	24.550	<0.001
Fat %	15.608	1.663	12.345	18.871	<0.001
NFC %	2.151	0.225	1.709	2.592	<0.001
Temperature (°C)	0.377	0.048	0.283	0.472	<0.001
Humidity (%)	0.155	0.022	0.113	0.197	<0.001
Milk yield (kg/day)	0.033	0.013	0.008	0.059	0.011
DIM (days)	0.005	0.001	0.004	0.007	<0.001
Humidity × Temperature	−0.007	0.001	−0.009	−0.005	<0.001
SCC × 1000	−0.0002	0.0001	−0.0004	−0.0001	0.006

CP—crude protein; NFC—non-fibre carbohydrates; Temperature—maximum daily temperature; Humidity—maximum daily humidity; DIM—days in milk; Humidity × Temperature—interaction term between maximum daily humidity and maximum daily temperature; SCC—somatic cell count; $\hat{\beta }$—estimate of the slope parameter; SE—standard error; CI—confidence interval.

**Table 3 animals-13-00295-t003:** Variance components analysis (from a generalized linear regression model) to assess the contribution of each variable to the variability of milk urea nitrogen (MUN) concentration. Data were collected from 161 Holstein cows from the Experimental Farm, University of Pretoria, during the period from February 2012 to November 2017.

Component	Variance Estimate	% Contribution
Other factors (error term)	6.148	23.8
CP %	5.406	20.9
Temperature (°C)	4.619	17.9
Humidity (%)	2.256	8.7
DIM (days)	2.033	7.9
Cow ID	1.937	7.5
SCC × 1000	1.693	6.6
NFC %	0.958	3.7
Milk yield (kg/day)	0.793	3.1

CP—crude protein; Temperature—maximum daily temperature; Humidity—maximum daily relative humidity; DIM—days in milk; Cow ID—contribution due to individual cow factors; NFC—non-fibre carbohydrates; SCC—somatic cell count.

## Data Availability

The data presented in this study are available on request from the corresponding author. The data are not publicly available due to protection of the client’s privacy.

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
