# Peer review of "Effect of Temperature and Humidity on Milk Urea Nitrogen Concentration"

_animals, 2023, doi:10.3390/ani13020295_

Round 1
Reviewer 1 Report
In this manuscript, the authors found that maximum daily ambient temperature and relative humidity had a significant influence on measured urea nitrogen concentration. The experiment was well designed, and the results presented by the authors are interesting. The paper shows evidence that supports the conclusions, however, the following aspects should be addressed.
Major Concerns:
1. Why did the author choose the commercially prepared lactation diets from three feed companies?Would feeding different diets affect the results?
2. As mentioned in the results,Half of the cows (54%) were followed only into the second parity, while 27%, 7% and 2% were present in the herd up to the third, fourth and fifth parity respectively. Then, does the author take into account whether different parity has an impact on MUN?
Minor Concerns:
1. The title should reflect how temperature and humidity affect the concentration of urea nitrogen in milk.
2. You should unify the page number format of references.
3. Line 183, 273 and 282: There should be a space between numbers and units.
Reviewer 3 Report
I was pleased to read this well-written paper.
Some minor changes should be taken into consideration:
1) the order of the table's variables: I would prefer to have in all tables the same arrangement of the variables, so that the first line is always the variable with the most important effect (or the most popular effect). Actually, the arrangement of the variables is not consistent (concerning Table 1, 2, and 3).
in Table 3, the variable (component) "fat" ist missing - is this correct?
2) lucerne component in the diet: You mention in the discussion the lucerne component in the diet. I think it would be fine to mention the roughage-compoment of the diet also in the part 2.1 (data collection and animal management).
3) model's variables: In line 149, it is mentinoned that nine variables were used in the model. I was wondering how many variables were used in the first step, before some were excluded. May be a separate list would help the reader to identifiy at a glance the excluded variables?
And how did you manage the corresponding (correlating) variables? (e.g. age and parity; milk conductivity and SCC, which doubtless will correlate with one another).
4) the consistent use of terms: in some cases, I had the impression that a more consistent use of the terms would be helpful.
"heat stress level" (L. 119) - I suppose you mean the THI-index?
SCC (Table 1 and 2) and "somatic cell count" (Table 3).
Why do you report SCC in log-scale? (Table 1, and likely also in Table 2 and 3, but there it is not marked). You calculated it in log, surely, but a median would be of interest to know, too (Table 1).
5) Table 2 and 3, the header: in Line 158, the model is named "multivariable linear mixed-effects regression model", and in Table 3, it is namend "generalized linear regression model" - is this a mistake?
6) font type and format:
- in the Tables, the font is sometimes bold in the first column, sometimes not.
- Line 10 to 11: the line-break is not correct
- Line 6: in the email-address, you should avoid a seperating dash (di-etmar...)
7) Minor spell / format remarks:
- Line 167: you write "unmeasured variables OR the error term of the model". I would think it should be "unmeasured variables (24%), REFLECTED IN the error term of the model" (or something like this - the word "or" has a more additive character, not an explaining one).
- Line 154: a comma is missing after "fat (p<0.001)"
